# A randomised, controlled, double blind study to assess mechanistic effects of combination therapy of dapagliflozin with exenatide QW versus dapagliflozin alone in obese patients with type 2 diabetes mellitus (RESILIENT): study protocol

Emily Brown ,[1,2] Moon M Wilton,[3] Victoria S Sprung ,[4] Joanne A Harrold,[3] Jason C G Halford,[5] Andrej Stancak,[3] Malcolm Burgess,[6] Elaine Howarth,[7] A Margot Umpleby,[8] Graham J Kemp,[9,10] John PH Wilding,[1,2] Daniel J Cuthbertson[2,3]

**Correspondence to**
Dr Emily Brown;
c.e.brown@liverpool.ac.uk

## ABSTRACT

**Introduction** The newer glucose-lowering therapies for type 2 diabetes (T2D), the glucagon-like peptide-1 receptor agonists (GLP1-RAs) and the sodium-glucose co-transporter 2 inhibitors (SGLT2i), have additional clinical benefits beyond improving glycaemic control; promoting weight loss, addressing associated cardiovascular risk factors and reducing macrovascular and microvascular complications. Considering their independent mechanisms of actions, there is a potential for significant synergy with combination therapy, yet limited data exist. This 32-week randomised, double-blind, placebo-controlled trial will gain mechanistic insight into the effects of coadministration of exenatide QW, a weekly subcutaneous GLP1-RA, with dapagliflozin, a once daily oral SGLT2i, on the dynamic, adaptive changes in energy balance, total, regional and organ-specific fat mass and multiorgan insulin sensitivity.

**Methods and analysis** 110 obese patients with diagnosed T2D (glycated haemoglobin, $HbA_{1c}$ ≥48 mmol/mol) will be treated for 32 weeks with dapagliflozin (10 mg once daily either alone or in combination with exenatide QW (2 mg once weekly); active treatments will be compared with a control group (placebo tablet and sham injection). The primary objective of the study is to compare the adjusted mean reduction in total body fat mass (determined by dual-energy X-ray absorptiometry, DEXA) from baseline following 32 weeks of treatment with exenatide QW and dapagliflozin versus dapagliflozin alone compared with control (placebo). Secondary outcome measures include changes in (1) *energy balance* (energy intake and energy expenditure measured by indirect calorimetry); (2) *appetite* (between and within meals) and satiety quotient; (3) *body composition* including visceral adipose tissue, subcutaneous adipose tissue, liver and pancreatic fat. Exploratory outcome measures include *metabolic changes* in hepatic and peripheral

## Strengths and limitations of this study

► This is the first study to assess the effects of sodium-glucose co-transporter 2 inhibitor (SGLT2i) and glucagon-like peptide-1 receptor agonist (GLP1-RA) combination therapy on changes in body composition and appetite in a randomised controlled trial design incorporating short-term and long-term effects.

► We employ an array of experimental techniques to identify the underlying biological basis of these changes and to yield some mechanistic insight into the metabolic, cardiac and vascular effects of combination therapy.

► The study is not designed to examine the individual effects of each agent but rather to investigate the compensatory mechanisms that additional GLP-1 RA therapy to SGLT2i therapy may induce.

► The relatively small sample size provides sufficient power to answer the primary research question.

► The relatively low sensitivity of even '*state-of-the-art*' research methods for detecting changes in energy intake, expenditure and energy balance mean that subtle changes may be missed.

insulin sensitivity (using a two-stage hyperinsulinaemic, euglycaemic clamp), *central nervous system responses* to food images using blood oxygen level-dependent (BOLD) functional MRI (fMRI) and *changes in cardiovascular function* (using transthoracic echocardiography, cardiac MR and duplex ultrasonography).

**Ethics and dissemination** This study has been approved by the North West Liverpool Central Research Ethics Committee (14/NW/1147) and is conducted in accordance with the Declaration of Helsinki and the Good Clinical

Practice. Results from the study will be published in peer-reviewed scientific and open access journals and/or presented at scientific conferences and summarised for distribution to the participants.

**Trial sponsor** University of Liverpool.

**Trial registration number** ISRCTN 52028580; EUDRACT number 2015-005242-60.

## BACKGROUND AND RATIONALE

Type 2 diabetes (T2D) is closely related to obesity: their prevalence is rising in parallel, synergistically accelerating and increasing the severity of diabetes-related complications.[1] Weight loss is therefore a primary therapeutic target in the management of patients with T2D. While some glucose-lowering treatments are associated with weight gain (eg, sulfonylureas, thiazolidinediones and insulin), two recent classes of therapeutic agents for T2D, namely, the glucagon-like peptide-1 receptor agonists (GLP1-RAs) and the sodium-glucose co-transporter 2 inhibitors (SGLT2i), are associated with weight loss, achieved via different mechanisms: increased urinary glucose loss for SGLT2i, and promotion of satiety for GLP-1 RAs.

Pharmacological inhibition of SGLT2 *reduces renal glucose reabsorption* in an insulin-independent mechanism promoting a daily urinary glucose excretion (UGE) of ~75 g, an energy deficit of ~300 kcal and a diuresis of ~400 mL.[2 3] GLP1-RAs confer similar therapeutic benefits operating via distinct mechanisms: *improving glycaemic control* (by enhancing glucose-dependent insulin secretion and inhibiting hepatic glucose production)[4] and *reducing body weight* via a central effect (promoting satiety via the hypothalamus).[5 6] The glucose-lowering effects of SGLTi maybe attenuated by increased endogenous (hepatic) glucose production,[7 8] partially explained by a compensatory hyperglucagonaemia.[9] It is also unclear why the magnitude of weight loss with SGLT2i therapy is not commensurate with the expected weight loss based on the determined caloric loss, suggesting compensatory or adaptative changes in energy intake and energy expenditure that attenuate the energy imbalance.[10 11]

Overall, the limited available evidence demonstrates the additive benefits of combination therapy with a GLP-1 RA and SGLT2 inhibitor on glycaemic improvement and weight.[12–14] Further studies are needed to elucidate their combined effects on metabolic and cardiovascular disease. The central hypothesis of this mechanistic study is that coadministration of exenatide once weekly (QW; a GLP-1 RA) with dapagliflozin (an SGLT2i) treatment will offset the potentially (mal)adaptive compensatory metabolic and appetitive responses occurring with dapagliflozin alone. Specifically, we anticipate that GLP-1 RAs can attenuate the increased hepatic glucose production due to hyperglucagonaemia and possible compensatory increases in appetite that arise with SGLT2i therapy (figure 1).

## PRIMARY OBJECTIVE

The primary objective of the study is to compare the adjusted mean reduction in total body fat mass (determined by dual-energy X-ray absorptiometry, DEXA) from baseline following 32 weeks of treatment with exenatide QW and dapagliflozin versus dapagliflozin alone compared with control (placebo).

## SECONDARY OBJECTIVES

The key secondary objectives are to assess the adjusted mean change from baseline of:
1. Metabolic measures
    – HbA$_{1c}$ (mmol/mol).
    – Fasting glucose (mmol/L).
    – Total Low-density lipoproteins (LDL) and High-density lipoproteins (HDL) cholesterol and triglycerides (mmol/L).

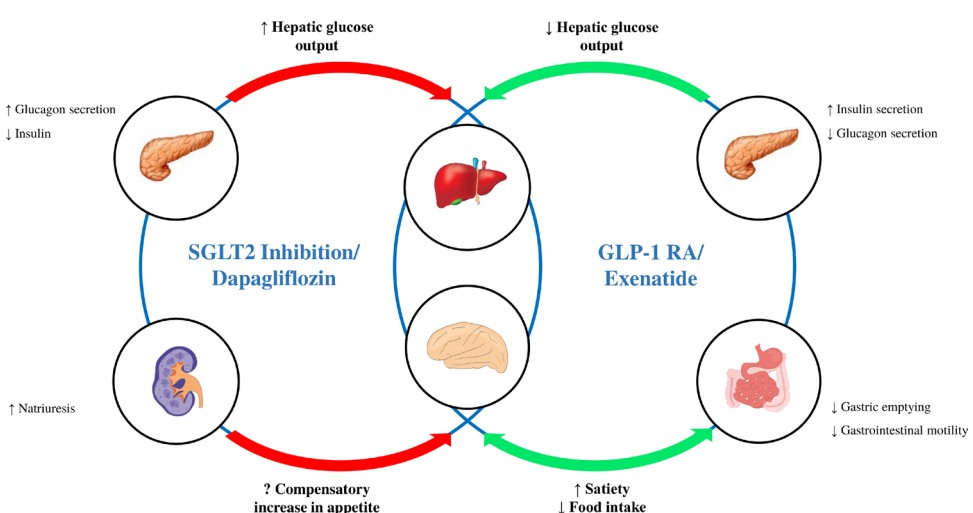

**Figure 1** Potential synergy with sodium-glucose co-transporter 2 (SGLT2) inhibitors and glucagon-like peptide-1 (GLP-1) receptor agonists.

– Twenty-four hour urinary glucose excretion (UGE) (g/dL).
2. Changes in energy balance
   – Food intake (g and kcal) at *ad libitum* meals.
   – Total energy expenditure: Measured by indirect calorimetry using a ventilated hood system and derived using the Weir equation.[15 16]
   – Physical activity: Measured by a multisensor array (daily average step count, and time spent in domains of physical activity (METS)).
3. Eating behaviour
   – Visual Analogue Scale (VAS): A seven-component VAS questionnaire used to assess hunger, fullness, prospective food consumption and desire to eat.
   – Satiety quotient: Measured using VAS ratings and calculated as the difference in hunger pre-meal and post-meal divided by the within meal caloric intake.
   – Psychology of appetite questionnaire scores.
4. Changes in body composition
   – Reduction in total body weight (kg).
   – Total fat volume (L), regional fat volumes (visceral fat and subcutaneous fat volume) and tissue-specific fat content (liver and pancreatic fat) measured by MRI scanning.
   – Changes in lean body mass measured by DEXA and MRI.
5. Changes in cardiovascular variables
   – Blood pressure (mm Hg).

In addition, this study will assess treatment compliance at 28-day intervals, based on the number of injections administered and returned tablet counts.

## EXPLORATORY ENDPOINTS

These are to be analysed and presented separately to primary and secondary outcome measures.
1. Metabolic measures
   – Endogenous (hepatic) glucose production, during a euglycaemic low-dose insulin clamp combined with a constant infusion of $[6,6\ ^2H_2]$ glucose.
   – Peripheral insulin sensitivity, measured by glucose uptake (Rd) and glucose metabolic clearance rate (MCR), during a euglycaemic high-dose insulin clamp combined with a constant infusion of $[6,6\ ^2H_2]$ glucose.
2. Changes in energy balance
   – Total energy expenditure: Measured by doubly labelled water.
3. Changes in cardiovascular variables:
   – Echocardiographic derived measures of left and right ventricular structure and function, including systolic and diastolic strain (rate) and twist mechanics.
   – Carotid artery intima media thickness (cIMT; mm) and brachial artery flow-mediated dilatation (FMD) using duplex ultrasonography.
   – MRI derived measures of cardiac structure and function.

4. Changes in body composition
   – Liver fibrosis measured by multiparametric MRI scanning.
5. Changes in Central nervous system (CNS) response
   – Differences in neuronal activity in CNS reward and satiety circuits (including striatum, amygdala, orbitofrontal cortex, insula, hypothalamus) as represented by BOLD (blood oxygen level-dependent) functional MRI (fMRI) signal change.

## SAFETY OBJECTIVE

The type and frequency of adverse events will be reported for all treatment groups.

## METHODS AND ANALYSIS
### Overall design, investigational plan and study population

This is a 32-week outpatient, double-blind, parallel group, randomised placebo-controlled mechanistic study of exenatide QW (GLP1-RA) and dapagliflozin (SGLT2i) treatment. One hundred and ten participants with obesity and T2D will be recruited: male and female, aged 18–65 years, Body mass index (BMI) 30–50 kg/m$^2$, with glycated haemoglobin (HbA$_{1c}$) ≥6.5% but ≤11% (48–97 mmol/mol; see table 1 for complete criteria). Each participant will have 13 study visits. Participants will be randomised equally to each of the three treatment arms (figure 2).

### Patient and public involvement

Patients were involved in the design and conduct of the trial. During the feasibility stage, choice of outcome measures, the number of study visits and methods of recruitment were informed through discussions with patients in the clinical setting and attendance at local diabetes education sessions. During the trial, a patient joined the independent trial steering committee (TSC). Once the trial has been published, a clear 'plain English' summary of the findings will be created so that they are widely available to participants and the wider patient groups. These will be presented to attendees at diabetes awareness days at the hospital and within the community.

### Sample size

This trial will examine exenatide QW, a weekly GLP1-RA, added to dapagliflozin versus sham saline injection added to dapagliflozin for an intervention period, of 32 weeks compared with a control group (placebo tablet and sham injection).
▶ **Arm A:** Placebo tablet once daily and sham injection once weekly.
▶ **Arm B:** SGLT2 (dapagliflozin 10 mg once daily plus sham injection once weekly).
▶ **Arm C:** SGLT2+GLP1 RA (dapagliflozin 10 mg plus exenatide 2 mg once weekly injection).

The sample sizes have been calculated in order to conduct the following two comparisons:
i.   Arm B (dapagliflozin) vs A (placebo).

**Table 1** Study inclusion and exclusion criteria

**INCLUSION CRITERIA**
► Males or females, age 18–65 years.
► A clinical diagnosis of type 2 diabetes.
► Glycosylated haemoglobin (HbA1c) ≥6.5% but ≤11% (48–97 mmol/mol).
► Currently treated with either diet or any combination of metformin, DPP-IV inhibitors* and sulfonylureas (excluding patients treated with pioglitazone or insulin). *DPP-IV inhibitors will require wash out period of 4 weeks.
► BMI 30–50 kg/m$^2$.
► Patients who are receiving the following medications must be on stable treatment regimen for a minimum of 2 months prior to screening:
 – Thyroid hormone replacement
 – Antidepressants

**EXCLUSION CRITERIA**
**Medical History and Concurrent Diseases**
► Type 1 diabetes mellitus.
► History of diabetic ketoacidosis or hyperosmolar non-ketotic coma.
► Renal impairment: eGFR less than 60 mL/min/1.73 m$^2$.
► Familial renal glucosuria.
► Clinically significant abnormal free T4 or patients needing initiation or adjustment of thyroid treatment according to the investigator.
► Severe uncontrolled hypertension ≥180 mm Hg and/or diastolic ≥110 mm Hg.
► Congestive heart failure class III-IV.
► Recent (<6 months) myocardial infarction.
► Known chronic liver disease (other than hepatic steatosis).
► Significant cardiac dysrhythmias (including pacemaker or ICD).
► Previous stroke.
► History of seizures or unexplained syncope.
► History of, or currently have, acute or chronic pancreatitis.
► History of malignancy (with the exception of basal and squamous cell carcinoma of the skin) within the last 5 years.
► History of medullary thyroid carcinoma or MEN2 (Multiple endocrine neoplasia type 2) or family history of medullary thyroid carcinoma or MEN2.
► History of gastric bypass surgery or gastric banding surgery, or either procedure is planned during the time period of the study.
► Patient, who in the judgement of the investigator, may be at risk of dehydration or volume depletion (hypovolaemia) that may affect the patient's safety and/or the interpretation of efficacy or safety data.
► Presence of any other medical condition that would, in the opinion of the investigator or their clinician, preclude safe participation in the study. This decision should be informed by dapagliflozin and exenatide precautions for use statements which will be provided to all clinicians and the research team.
► Alcohol consumption in excess of 21 units/week females, 28 units/week males.
► Current smoker (including electronic cigarettes) or having ceased smoking in the last 6 months.
► Any history of a pacemaker or implantable cardioverter defibrillator (ICD).
► Patients with a history of diabetic foot ulcers or previous (lower limb) digital amputations.

► Patients currently taking part in a structured weight loss programme such as weight watchers or slimming world (excluding those on weight management programmes as part of their diabetes care).
► Patients who score over 4 restraint subscale of the Dutch Eating Behaviour Questionnaire.
► Patients who score over 27 on the Binge Eating Scale.
**Physical and Laboratory Test Findings**
► ALT>3×upper limit of normal (ULN).
► AST>3×ULN.
► Bilirubin>2×ULN.
► Haemoglobin ≤10.5 g/dL (≤105 g/L) for men; haemoglobin ≤9.5 g/dL (≤95 g/L) for women.
► History of unexplained microscopic or macroscopic haematuria at screening, confirmed by follow-up sample a next scheduled visit, where according to the investigator a satisfactory evaluation of haematuria has not been conducted.
► Weight <60 kg and >200 kg (due to DEXA limitations).
► BMI <30 kg/m$^2$ and >50 kg/m$^2$.
► Recent major change in body weight (>3 kg loss or gain in preceding month).
**Allergies and Adverse Drug Reactions**
► Subjects with a history of any serious hypersensitivity reaction to GLP1-RA or SGLT2 inhibitor.
► Participant should have no allergies against metacresol (the preservative in insulin vial).
► History of anaphylaxis to food.
► Known food allergies or food intolerance.
► Known hypersensitivity to heparin.
► Known hypersensitivity to intravenous catheter equipment.
**Sex and Reproductive Status**
► Females of childbearing age who are not using adequate contraceptive methods or who are planning a pregnancy in the next 44 weeks (study duration plus 12 weeks).
► Women who are pregnant or breast feeding.
► Sexually active fertile men not using effective birth control if their partners are WOCBP.
**Prohibited Treatments and/or Therapies**
► Diabetes treated with pioglitazone, SGLT2 inhibitors, GLP-1 analogues or insulin.
► Treatment with SGLT2 inhibitor, GLP1 RA or subcutaneous insulin injections 3 months before screening.
► Any weight loss medication (eg, orlistat) within 3 months prior to screening.
► Use of any drug that might affect body weight or appetite (including antipsychotics or corticosteroids) within 3 months prior to screening.
► Patients who are currently receiving a loop diuretic that cannot be discontinued.
**Other Exclusion Criteria**
► Active or previous substance abuse or dependence.
► Prisoners or subjects who are involuntarily incarcerated.
► Subjects who are compulsorily detained for treatment of either a psychiatric or physical (eg, infectious disease) illness.
► Dislike >25% of the study foods.
► Participation in other studies (within the past 30 days).
*In situations where the patient has a contraindication to MR scanning, they will be randomised to the study but excluded from MR scanning only. The primary outcome measurement of fat mass can be derived from DEXA.*

ALT, Alanine transaminase; AST, Aspartate transaminase; BMI, Body mass index; DEXA, dual-energy X-ray absorptiometry; eGFR, estimated glomerular filtration rate; GLP1-RA, glucagon-like peptide-1 receptor agonist; SGLT2i, sodium-glucose co-transporter 2 inhibitors; WOCBP, Women of childbearing potential.

ii. Arm B (dapagliflozin) vs C (dapagliflozin and exenatide).

These comparisons are selected to answer the key question of interest; although the study will not provide comparisons between arms C and A, nor will they be reported.

The original sample size calculation was carried out using Stata V.14 software (Stata Statistical Software: Release 14. College Station, Texas, USA: StataCorp LP). For both comparisons, a 1.5 kg in the change in body fat mass is considered a clinically relevant difference. From previous data, the within patient SD for the change in total body fat between baseline and 32 weeks is estimated as 1.8 kg.[17]

From this, a between-patient SD of 2.55 kg was computed. Using a two-group test with equal means (assuming 80% power and one-sided 5% significance level), 36 patients per treatment arm were estimated to be required in order to adequately power the planned analysis (assuming effect size of 0.59). Using these calculations and an anticipated 10% attrition and patient dropout rate, the original sample size was set at 120 participants.

As a result of the COVID-19 pandemic, however, 15 active patients were lost to follow-up and a revised sample size calculation was implemented, reducing the original sample size from 120 patients to 110 randomised patients in order to maintain meaningful results within the

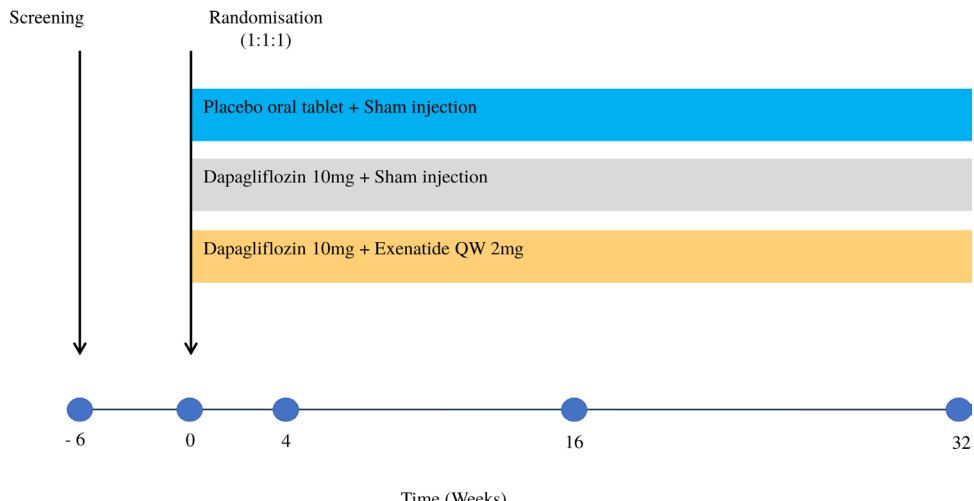

**Figure 2** Study schematic.

remaining timeframe of the study. Using the same clinically relevant difference of 1.5 kg, significance level and power, the between-patient SD would be 2.42 kg, giving an effect size of 0.62. Assuming 80% power, a minimum of 33 patients per treatment arm are required. Allowing for an anticipated 11% attrition rate/dropout rate, a total of 110 patients are required (calculated using NQuery V.8.1.2.0).

Recruitment will take place over 44 months in Liverpool University Hospital NHS Foundation Trust (Aintree site), Liverpool, UK. Participants will be recruited either from existing databases of volunteer patients, diabetes clinics in the hospital and community and by advertisement in the local press and social media.

### Screening, enrolment and randomisation

After giving written informed consent, potentially eligible participants will attend a screening visit within 6 weeks prior to randomisation; this includes a medical history to confirm the participant's eligibility to participate as determined by the inclusion/exclusion criteria (table 1), physical examination, blood tests, urinalysis and an ECG. Each participant will be informed of his/her eligibility for the trial once all results are available (usually within 1 week from obtaining consent).

The randomisation for each stratum will be performed using computer generated (Treatment Allocation Randomisation System, TARDIS) random balanced blocks to ensure approximately equal numbers of participants across the treatment arms within each stratum. Each patient will be assigned a unique study identification number. The study is set-up as a double blind trial with both the participants, doctor, site and LCTC (Liverpool Clinical Trials Centre) RESILIENT trial teams being blind to the treatment received. Unblinded members of the research staff (pharmacy and selected research nurses) will remain the same throughout the trial. A blinded confirmation email will be sent to all relevant site staff and LCTC trial team detailing the trial number and patient initials. An unblinded confirmation email will also be

sent to all relevant pharmacy staff detailing the allocated treatment arm, trial number and patient initials. Only the subject number and initials will be recorded in the electronic case report form (eCRF). All other patient identifiable data will be completely anonymised. An additional validation check has been built into the eCRF (MACRO 4) for authorised trial team members only to confirm eligibility and suitability to proceed to randomisation.

### Dosage and administration of study treatments

Participants will attend once a week for administration of a subcutaneous injection (exenatide QW 2 mg or sham injection) for the duration of the study (32 weeks). The exenatide QW injection and the sham injection are not identical in appearance, so to maintain the study blinding, medication dispensed will be given to a member of the unblinded research team in an opaque bag to ensure the contents cannot be seen by members of the blinded research team or the patient. No blinded staff member will be allowed in the room at the time of injection and the injection will be disposed of by unblinded staff members in line with local protocols, and in a concealed manner. Participants will also be instructed to take a tablet with water (dapagliflozin 10 mg or matching placebo) each morning for the duration of the study, while continuing with their usual medication and attending for scheduled study visits. The number and size of tablets will be identical for the investigational products (dapagliflozin 10 mg and placebo). Participants will be asked to return all unused investigational products, including any empty medication packages to the clinic at each visit. The subject's compliance will be discussed at each study visit and assessed based on the number of injections administered and returned counts. This will be recorded in the defined CRF. To aid compliance, participants will be asked to complete a patient diary. Participants judged to be non-compliant may continue in the study, but will be counselled on the importance of taking their study medication as prescribed.

## Study visits and procedures

### Anthropometric measurements

Body weight and height, waist and hip circumference and whole-body bioimpedance analysis (TANITA: total body mass, fat percentage, fat mass, fat free mass and muscle mass) will be recorded. Blood pressure will be measured from the left arm after patients have been seated for 5 min. The first reading will be discarded, and the mean of three subsequent readings recorded (table 2).

### Biochemical measurements

All patients will have a routine blood sample taken for $HbA_{1c}$, fasting plasma glucose, renal profile (to include electrolytes, urea and creatinine, and eGFR), lipid profile, liver function tests (to include ALT, AST and bilirubin) and insulin. Insulin sensitivity will be measured by Homeostatic Model Assessment for Insulin Resistance (HOMA-IR).[18] eGFR will be estimated using the Modification of Diet in Renal Disease (MDRD). We will measure fasting plasma glucose (FPG) and estimated glomerular filtration rate (eGFR) every 2–4 weeks. This will allow us to determine the amount of filtered glucose calculated from the product of the FPG (mg/dL) and eGFR (mL/min/1.73 m²).

### Test meal visits (visits 1, 4, 7 and 11)

Test meal visits will take place at baseline and after 4, 16 and 32 weeks of treatment (tables 2 and 3). Participants will be asked to attend the investigational unit at 08:00, having had nothing to eat or drink other than water from midnight. Participants' anthropological and biochemical measurements will be taken (see above). An explanation and demonstration of Visual Analogue Scale (VAS) questionnaires, appetite questionnaires and ventilated hood will be given. Additional measures taken during test meal days are described below.

### Food intake and eating behaviour

Participants will be provided with a fixed load breakfast (providing 25% energy from estimated Resting metabolic rate (RMR)) followed 4 hours later by an *ad libitum* lunch, which they will self-serve. The *ad libitum* lunch will consist of a multiple item buffet (eg, high and low calorie, sweet and savoury items) to understand how the potential satiety and reward-based changes in appetite effect energy intake and modify food choice. Pen and paper VAS measures of appetite (hunger, fullness, desire to eat, prospective consumption, satisfaction, thirst and nausea) fluctuations throughout the day will also be collected with retrospective appetite and craving questionnaires at the end of the day. Taken together, these data fully characterise drug effects on appetite and eating behaviour.

A range of questionnaires assessing secondary outcomes of appetite, cravings, feelings of control and eating behaviours (including food choice) will also be completed at test meal visits. These include the Three Factor Eating Questionnaire (TFEQ)[19]; Power of Food Scale (PoF)[20]; the Craving for Sweet Foods sub-questionnaire from the

Health and Tastes Attitude Scale (HTAS)[21]; the 7-day Control of Eating Questionnaire (COEQ)[22]; the external eating subscale from the Dutch Eating Behaviour Questionnaire (DEBQ)[23] assessing external eating; and the Mindful Eating Scale (MES)[24] describing the physical and emotional sensations associated with eating.

### Indirect calorimetry

Energy expenditure and respiratory quotients will be measured on the morning of the test meal visit prior to the fixed energy breakfast using an indirect calorimeter (GEM Nutrition, Daresbury, UK) fitted with a ventilated hood and derived using the modified Weir equation.[16] Participants will remain awake but motionless in a supine position for 45 min with RMR calculated from respiratory data averaged over the final 30 min of assessment. The first 15 min of each measurement will be discarded to allow for complete acclimatisation to the hood and the recumbent position.

### Physical activity monitoring

Habitual physical activity will be monitored using a Sense-Wear mini armband (BodyMedia, Pittsburgh, Pennsylvania, USA) for a 4-day period (to include one weekend day) during weeks 0, 4, 16 and 32. Data collected from the SenseWear device include daily average step count, total energy expenditure, active energy expenditure and time spent in domains of physical activity including sleep, lying, sedentary (<1.5 metabolic equivalents, METS), light (1.5–3 METS), moderate (3–6 METS), vigorous (6–9 METS) and very vigorous (>9 METS) and is analysed using SenseWear Professional software (V.8.0).

### Food diaries

Participants will be asked to complete a food diary detailing exactly what they eat and drink over the same 4 day period. Total energy consumption, carbohydrate, protein and fat content will be determined from dietary records using Nutrics (Nutrition Analysis Software for Professionals).

## Baseline and end of treatment measures

### Body composition

All participants will undertake a full body DEXA (Lunar IDXA, GE Healthcare, Amersham, UK) scan after an overnight fast at baseline and week 32. Body composition will be determined to quantify total and regional (trunk, limb, android and gynoid) fat mass and fat-free mass.

MRI measures of body composition, hepatic and pancreatic fat MRI eligible participants will be scanned using a 3.0 T Siemens Prisma scanner (Siemens Medical Solutions, Erlangen, Germany) at the Liverpool MRI Centre (LiMRIC). Images are obtained with the patient supine and using an integrated body coil (Siemens Medical Solutions, Erlangen, Germany).

For body composition, multiple blocks of contiguous slices using a Dixon VIBE in and out of phase sequence will be conducted to produce fat and water images from the vertex of the skull to the knees. These will be used to

**Table 2** Assessments and procedures during screening and study visits (D, day; W, week)

\* Not including administration of weekly exenatide QW/sham injection

| Procedure | D −42–2 (including washout if necessary) | D −14–0 | D 0 | W 2 | W 4 | W 8 | W 12 | W 16 | W 20 | W 24 | W 28 | W 30–32 |
|---|---|---|---|---|---|---|---|---|---|---|---|---|
| Visit (fasting) | 0 | | 1 and 2 | 3 | 4 | 5 | 6 | 7 | 8 | 9 | 10 | 11 and 12 |
| Informed consent | X | | | | | | | | | | | |
| Medical history/ demographics | X | | | | | | | | | | | |
| Physical examination and ECG | X | | | | | | | | | | | |
| Pregnancy test | X | | X | | | | | | | | | |
| Glucometer dispensing and training | X | | | | | | | | | | | |
| Urinalysis | X | | | | | | | | | | | |
| Height | X | | | | | | | | | | | |
| Weight | X | | X | X | X | X | X | X | X | X | X | X |
| Waist and hip measurements | | | X | X | X | X | X | X | X | X | X | X |
| Bioelectrical impedance analysis (BIA) | | | X | X | X | X | X | X | X | X | X | X |
| BP, pulse | X | | X | X | X | X | X | X | X | X | X | X |
| Blood tests | X | | X | X | X | X | X | X | X | X | X | X |
| Dapaglifozin/placebo dispensing | | | X | | X | X | X | X | X | X | X | X |
| Concomitant medication check | X | | X | X | X | X | X | X | X | X | X | X |
| Compliance check | | | | X | X | X | X | X | X | X | X | X |
| AE reporting | | | X | X | X | X | X | X | X | X | X | X |
| 24-hour urine | | | X | | X | | | X | | | | X |
| 4-day food diary | | | X | | X | | | X | | | | X |
| 4-day activity monitoring | | | X | | X | | | X | | | | X |
| Test meal | | | X | | X | | | X | | | | X |
| Indirect calorimetry | | | X | | X | | | X | | | | X |
| Psychological questionnaires and VAS | | | X | | X | | | X | | | | X |
| DEXA | | | X | | | | | | | | | X |
| MRI (full body, liver fat, pancreatic fat) | | | X | | | | | | | | | X |
| MRI cardiac | | | X | | | | | | | | | X |
| fMRI and eye-tracking | | X | | | | | | | | | | X |
| FMD | | | X | | | | | | | | | X |
| Cardiac ECHO | | | X | | | | | | | | | X |
| DLW and additional DEXA | | X | X | | X | | | X | | | | X |
| Euglycaemic clamps | | | X | | | | | | | | | X |

*Not including administration of weekly exenatide QW/sham injection.
AE, adverse event; BIA, bioelectrical impedance analysis; BP, blood pressure; DEXA, dual-energy X-ray absorptiometry; DLW, doubly labelled water; ECG, electrocardiogram; FMD, flow-mediated dilatation; fMRI, functional MRI; VAS, Visual Analogue Scale.

**Table 3** Test meal guidelines

| Time point guidelines | Procedure |
|---|---|
| Arrival 08:00* | Blood sample taken.<br>Subjects asked to empty bladder; start of timed 24-hour urine collection.<br>Weight, waist circumference, bioelectrical impedance analysis, pulse and blood pressure taken. |
| 08:10 | Basal metabolic rate measurement (indirect calorimetry—see below for details). |
| 08:55 (T1 VAS) | Subjects will complete a pre-breakfast, seven-component VAS questionnaire to measure hunger and other subjective variables. |
| 09:00 | Subjects will be provided with a fixed-quantity breakfast, consisting of cornflakes with milk, toast and preserve, tea/coffee and orange juice. |
| 09:20 (T2 VAS) | Subjects will complete a post-breakfast, 7-component VAS questionnaire to measure hunger and other subjective variables. |
| 10:00, 11:00, 12:00 (T3-5 VAS)<br>*Administered at hourly intervals (from pre-breakfast VAS) for 3 hours* | Subjects will complete the same seven-component VAS questionnaire as used pre-breakfast, providing a pre-meal set of ratings. |
| 12:55 (T6 VAS) | Subjects will complete a pre-lunch, seven-component VAS questionnaire to measure hunger and other subjective variables. |
| 13:00 | Subjects will be given an ad libitum lunch. The meal consists of a multiple item buffet (mix of high-fat and low-fat savoury items) and subjects may eat ad lib, and signal when they have finished the meal. |
| 13:30 (T7 VAS) | Subjects will complete a post-lunch, seven-component VAS questionnaire to measure hunger and other subjective variables.<br>Participants also complete a post-lunch palatability VAS. |
| 14:00, 15:00, 16:00, 17:00 (T8-11 VAS)<br>*Administered at hourly intervals (from pre-breakfast VAS) for 4 hours* | Subjects will complete the same seven-component VAS questionnaire as used pre-breakfast, providing a post-meal set of ratings. |

*Timings will be adjusted to facilitate an amended arrival time. All events will remain at the same relative intervals.
VAS, Visual Analogue Scale.

determine fat volumes in abdomen and neck. For hepatic fat, a multi-echo Dixon sequence with blocks of contiguous slices and breath holds on expiration will be used to image the abdomen. A single transverse slice that intersects the porta-hepatis will be positioned and a multi-echo gradient-echo image will be acquired to determine a liver fat fraction. For the assessment of pancreatic fat, circular regions of interest ($\sim$100 mm$^2$) will be drawn in the head, body and tail of the pancreas of two representative slices, with fat fraction calculated as the average across both slices.

### Exploratory outcomes

All participants will be offered the opportunity to take part in several optional study visits with a view to 20 participants completing each treatment ARM. Uptake to these optional visits will be determined by eligibility, specifically contraindications to MR scanning and patient choice.

A. *Euglycaemic–hyperinsulinaemic clamp:* A subset of participants will take part in a two-stage, low-dose and high-dose euglycaemic–hyperinsulinaemic clamp combined with a [6,6 $^2$H$_2$] glucose infusion, at baseline and week 32 (described in detail in Cuthbertson *et al*[25]). Briefly, a primed continuous infusion of [6,6

$^2$H$_2$] glucose will be administered intravenously for 6 hours. After 2 hours, insulin infusion at 0.5 mU·kg$^{-1}$·min$^{-1}$ (low-dose) will be administered for 2 hours to measure the insulin sensitivity of endogenous glucose production (EGP); then insulin infusion at 1.5 mU·kg$^{-1}$·min$^{-1}$ (high-dose) will be administered for 2 hours to measure insulin sensitivity of peripheral glucose uptake (Rd). Euglycaemia will be maintained by adjusting a 20% glucose infusion, spiked with glucose (7 mg·g$^{-1}$ for low dose, 10 mg·g$^{-1}$ for high dose) according to 5 min plasma glucose measurements using a glucose oxidase method (Yellow Springs Analyser). Blood samples will be obtained before the start of the tracer infusions, every 10 min during the final 30 min of the basal period, low-dose and high-dose steady state of the clamp procedure, and every 30 min between these periods. The isotopic enrichment of plasma glucose will be determined by gas chromatography mass spectrometry at the Wolfson Centre for Translational Research, Postgraduate Medical School, University of Surrey, UK.

B. *Doubly labelled water (DLW):* A subset of participants will undertake the doubly labelled water method com-

 Brown E, *et al. BMJ Open* 2021;**11**:e045663. doi:10.1136/bmjopen-2020-045663

bined with a DEXA scan, at weeks 0, 4, 16 and 32 to assess mean energy expenditure over a 2-week period. Participants will drink from a stock solution of $^2H_2O$ (99% enrichment) and $H_2^{18}O$ (10% enrichment) at a dose of 1 g·kg$^{-1}$. Spot urine samples will be collected at 1.5-hour intervals on the day of DLW administration, followed by once daily for 13 days. Isotopic enrichments of the urine samples will be measured by isotope ratio mass spectrometry at The National Institute of Diabetes and Digestive and Kidney Disease (NIDDK), Bethesda, USA.

C. *Endothelial function and vascular structure:* A 10 MHz multifrequency linear array probe attached to a high-resolution ultrasound machine (Siemens Medical Solutions, Malvern, Pennsylvania, USA) will be used to image the carotid artery and the brachial artery (in the distal third of the upper right arm) at baseline and week 32 in a subset of participants. Nitric oxide–mediated endothelial function will be assessed by measuring flow-mediated dilatation (FMD) in response to a 5 min ischaemic stimulus, induced by forearm cuff inflation.[26] Peak brachial artery diameter and blood flow velocity, and the time taken to reach these peaks following cuff release will be recorded. The structure of the carotid artery will be measured as the intima-media thickness (cIMT), or artery wall thickness using custom software.[27]

D. *Cardiac structure and function:* Indices of myocardial systolic and diastolic function will be determined using 2D and tissue Doppler echocardiography.[28] All echocardiograms will be performed using a GE E95 machine with an M5SC-D 2.5 MHz phased array transducer, measuring left ventricular (LV) dimensions during diastole and systole and wall thicknesses and LV mass; LV and ejection fraction (modified Simpson's biplane method); left atrial volumes; mitral inflow peak early diastolic velocity (E), peak late diastolic velocity (A), E/A ratio and E wave deceleration time. Tissue Doppler mitral annular early diastolic velocity (E′) will be assessed at septal and lateral sites, and averaged for calculation of E/E′. Global longitudinal strain will be derived from apical 4-chamber, apical 2-chamber and apical long axis images. All data will be analysed using Echopac (V.9: GE, Horten, Norway).

E. *Cardiac MRI:* A subset of participants will undertake cardiac magnetic resonance at baseline and week 32 using the same MRI scanner described combined with ECG gating. A gadolinium-based contrast agent (Gadovist, Bayer) will be administered via a cannula in the forearm to facilitate imaging. A 2.5 mL blood sample will be collected to measure the haematocrit. Strain and peak early diastolic strain rate will be quantified from tagging and cine images using InTag (www.creatis.insa-lyon.fr/inTag/) and feature tracking, respectively. LV volumes, mass and function will be calculated using commercially available non-proprietary software. Tissue characterisation will be performed using standard T1 mapping (correcting for haematocrit) and late gadolinium enhancement techniques.

F. *Functional Magnetic Resonance Imaging (fMRI):* Using the same MRI scanner, a subset of participants will undergo fMRI to examine neural processing in response to tasks that assess reward related function (Passive-Viewing) and inhibitory control (Stop-Signal Task).[29] Using the test-meal protocol described earlier, BOLD imaging will be conducted in patients under fasted (pre-lunch) and sated (post-lunch) states both at baseline and week 32. The fMRI data will be imported to and analysed using Statistical Parametric Mapping software (SPM12, University College London, UK) and custom built MATlab scripts (The MathWorks, Massachusetts, USA). Appetite will be measured throughout the scanning day using a VAS.

## Monitoring/dispensing visits (1, 4, 5, 6, 7, 8, 9, 10)

These visits will involve a brief consultation with the study team to review glycaemic control (self-monitored capillary blood glucose), adverse effects and compliance (tablet count), and to collect supply of medication. All injections of exenatide QW/placebo will be administered weekly by unblinded staff members at site and recorded. Unblinded members of the research team will have no role in assessment of participants outside of these specified roles as detailed in the trial delegation log.

## Pharmacovigilance

All adverse events will be reported and assignment of the severity/grading (mild, moderate, severe, life-threatening, death) made by the investigator responsible for the care of the participant. The assignment of causality will be made by the investigator. All non-serious adverse events (SAEs), whether expected or not, will be recorded and updated at each study visit. All new SAEs will be reported from the point of consent until 28 days after discontinuation of the investigational medical product; this includes those thought to be associated with protocol-specified procedures. Investigators will report SAEs, serious adverse reactions (SARs) and sudden unexpected adverse reactions (SUSARs) to LCTC within 24 hours of the local site becoming aware of the event. LCTC will notify the Medicines and Healthcare products Regulatory Agency (MHRA) and main Research Ethics Committee (REC) of all SUSARs occurring during the study: fatal and life-threatening SUSARs within 7 days of notification and non-life-threatening SUSARs within 15 days. All adverse events will be followed until satisfactory resolution or until the investigator responsible for the care of the participant deems the event to be chronic or the patient to be stable.

## Trial monitoring and oversight committees at LCTC

The RESILIENT study will have a Trial Management Group (TMG), Trial Steering Committee (TSC) and an Independent Safety and Data Monitoring Committee (ISDMC) to monitor study progress.

The TMG, supported by the LCTC, will be responsible for the day-to-day running and management of the trial and will consist of the protocol committee members and the trial manager. The TSC will provide oversight of the study, concentrating on progress of the study, adherence to protocol, participant's safety and consideration of new information, making recommendations on study pathway modifications and continuation of the study. The TSC will include experienced lay members of the public, other medical experts and clinical trialists. Meetings will be held at regular intervals determined by need, but no less than once a year. The ultimate decision for the continuation of the trial lies with the TSC.

The ISDMC will be responsible for reviewing and assessing recruitment, interim monitoring of safety and effectiveness, trial conduct and external data. The ISDMC will also provide recommendations to the TSC concerning continuation of the study.

## Outline of analysis
### General approach
Categorical variables will be summarised as frequency (%), and continuous variables will be summarised as mean (SD) or median (IQR) depending on the distribution. Full details of the planned statistical analysis will be included in a separate statistical analysis plan to be developed in the months following trial opening and will be approved by both the ISDMC and the TSC.

### *Subgroup analysis*
No subgroup analysis is planned for this trial.

### *Significance levels*
For the analysis of the primary outcome, statistical significance will be tested using Hochberg's procedure, a hierarchical analysis approach for the primary outcomes of (1) exenatide QW and dapagliflozin compared with placebo control, (2) dapagliflozin alone compared with placebo control. Here comparisons will be ordered (descending) in terms of their significance level and assessed using a p value $<0.05$ to determine statistical significance for the first comparison and $p<0.025$ for the second.

The main analysis for primary and secondary endpoints will use the full analysis set, consisting of all randomised patients, with participants analysed according to the group to which they were originally allocated, and with outcomes included irrespective of protocol adherence, in order to follow the *intention-to-treat* principle. The safety population, consisting of all patients who actually receive a trial intervention, according to the treatment received, will be used for analysis of toxicity and adverse events.

### *Missing data*
Missing data are assumed to be small and analysis will be carried out on a complete case basis. If there are a significant number of missing entries on any endpoint ($>20\%$ for example), missing data shall be estimated using multiple imputation based on the method of chained equations.

### Analysis
For analysis of the primary endpoint, the mean difference from baseline and week 32 will be presented with a corresponding 95% CI. Between-group differences will be analysed using linear regression techniques, analysing the week 32 values while including baseline covariate value. The effect of dapagliflozin will be estimated by contrasting (1) control arm versus dapagliflozin arm. The effect of exenatide will be estimated by comparing (2) dapagliflozin versus dapaglifozin and exenatide.

### *Secondary analyses*
Analyses of all secondary endpoints are continuous and analysis techniques shall replicate that of primary endpoints.

### Safety analysis
Information relating to adverse events (including events relating to hypoglycaemia, gastrointestinal upset, and urinary and genital tract infections) will be tabulated and summarised descriptively. Continuous laboratory values will be summarised as described above.

## ETHICS AND DISSEMINATION
This study is being conducted in accordance with Good Clinical Practice (GCP), as defined by the International Conference on Harmonisation (ICH) and in compliance with the European Union Directive 2001/20/EC transposed into UK law as statutory instrument 2004 No 1031: Medicines for Human Use (Clinical Trials) Regulations 2004 and all subsequent amendments and the US Code of Federal Regulations, Title 21, Part 50 (21CFR50). The trial protocol has received the favourable opinion of the NRES North West/Liverpool Central Research Ethics Committee (14/NW/1147; protocol number UoL 001187). An appropriate patient information sheet and consent forms describing in detail the trial interventions/products, trial procedures and risks were approved by the ethical committee (IEC), and the patients are asked to read and review the document. The investigator explains the study to the patient and answer any questions posed. A contact point where further information about the trial may be obtained is provided. After being given adequate time to consider the information, the patient is asked to sign the informed consent document. A copy of the informed consent document is given to the patient for their records and a copy placed in the medical records, with the original retained in the investigator site file. The patient may withdraw from the trial at any time by revoking the informed consent. The rights and welfare of the patients are protected by emphasising to them that the quality of medical care will not be adversely affected if they decline to participate in this study.

## REGULATORY APPROVAL
This trial has been registered with the MHRA and has been granted a Clinical Trial Authorisation (CTA). The CTA reference is its EudraCT number: 2013-004264-60.

## PUBLICATION

The results will be analysed together and published as soon as possible. The uniform requirements for manuscripts submitted to *biomedical journals* (http:// www. icmje.org/) will be respected. The ISRCTN allocated to this trial would be attached to any publications resulting from this trial.

## TRIAL STATUS

The trial opened for recruitment at University Hospital Aintree in Liverpool on 25 October 2017. Recruitment is ongoing at the time of publication with the anticipated trial completion (last patient last visit) in October 2021.

**Author affiliations**
[1]Metabolism and Nutrition Research Group, University Hospital Aintree, Liverpool University Hospitals NHS Foundation Trust, Liverpool, UK
[2]Department of Cardiovascular and Metabolic Medicine, Institute of Life Course and Medical Sciences, University of Liverpool, Liverpool, UK
[3]Department of Psychology, Institute of Population Health, University of Liverpool, Liverpool, UK
[4]Research Institute for Sport & Exercise Sciences, Liverpool John Moores University, Liverpool, UK
[5]School of Psychology, University of Leeds, Leeds, UK
[6]Department of Cardiology, University Hospital Aintree, Liverpool University Hospitals NHS Foundation Trust, Liverpool, UK
[7]Liverpool Clinical Trials Centre, Institute of Population Health, University of Liverpool, Liverpool, UK
[8]Department of Nutritional Sciences, University of Surrey, Guildford, UK
[9]Liverpool Magnetic Resonance Imaging Centre (LiMRIC), University of Liverpool, Liverpool, UK
[10]Department of Musculoskeletal and Ageing Science, Institute of Life Course and Medical Sciences, University of Liverpool, Liverpool, UK

**Correction notice** This article was previously published with wrong licence. The correct licence for the paper is CC-BY.

**Acknowledgements** We would like to thank Manoj Mistry for his support with the design of patient information sheets and for his feedback on the study design.

**Contributors** DJC, the principal investigator for this study, designed and wrote the original study protocol with JPHW, JAH and JCGH . VSS, JPHW, JCGH, JAH, AS, MB and GJK are coinvestigators for the study. EB and MMW are the clinical research fellow and post-doctoral associate responsible for overseeing the clinical trial and trial-related activities. EB drafted the first version of protocol manuscript in the appropriate journal format with subsequent input from all coauthors. JCGH and JAH developed the behavioural methodology for the protocol. AMU developed the euglycaemic-hyperinsulinaemic clamp methodology for the protocol. EH is the trial statistician, contributed to writing the manuscript and approved the final version. All authors have contributed to the revision of the manuscript and approved the final version.

**Funding** This work was funded by Astra Zeneca grant number ESR-14-10188.

**Competing interests** DJC has competing interests with AstraZeneca, Boehringer Ingelheim, Janssen Pharmaceuticals, Eli Lilly and Novo Nordisk. JPHW has acted as a consultant, received institutional grants and given lectures on behalf of pharmaceutical companies developing or marketing medicines used for the treatment of diabetes and obesity, specifically Astellas, AstraZeneca, Boehringer Ingelheim, Janssen Pharmaceuticals, Eli Lilly, Novo Nordisk, Napp, Mundipharma, Orexigen, Rhythm Pharmaceuticals, Sanofi and Takeda. EB and MMW are currently supported by a grant funded to the University of Liverpool by AstraZeneca.

**Patient and public involvement** Patients and/or the public were involved in the design, or conduct, or reporting or dissemination plans of this research. Refer to the Methods and analysis section for further details.

**Patient consent for publication** Not required.

**Provenance and peer review** Not commissioned; externally peer reviewed.

**ORCID iDs**
Emily Brown http://orcid.org/0000-0003-1097-0580
Victoria S Sprung http://orcid.org/0000-0002-2666-4986

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
