## [Reviewer comments · BMJ Open]

ARTICLE DETAILS

TITLE (PROVISIONAL)	A Randomised, controlled, double blind Study to assess mechanistic effects of combination therapy of dapagliflozin with Exenatide QW versus dapagliflozin alone in obese patients with Type 2 diabetes mellitus (RESILIENT) – study protocol
AUTHORS	Brown, Emily; Wilton, Moon; Sprung, Victoria; Harrold, Joanne; Halford, Jason; Stancak, Andrej; Burgess, Malcolm; Howarth, Elaine; Umpleby, A; Kemp, Graham J; Wilding, John PH; Cuthbertson, Daniel

VERSION 1 – REVIEW

REVIEWER	Nauck, M Ruhr-University Bochum, Division of Diabetology
REVIEW RETURNED	28-Oct-2020

GENERAL COMMENTS	Brown et al. present a protocol for a clinical study comparing dapagliflozin (SGLT-2 inhibitor), exenatide once weekly (GLP-1 receptor agonist) plus dapagliflozin, and placebo over 32 weeks, with the primary endpoint change in fat mass, and various secondary endpoints focusing on energy balance, appetite/satiety and body composition. Mechanistic questions will be examined in subgroups (insulin sensitivity, CNS processing of food-related clues, cardiovascular function, etc.). The question of how a combination of GLP-1 receptor agonists and SGLT-2 inhibitors works, is of high interest, mainly with respect to cardiovascular outcomes (which will not be clarified with the present protocol). The main deficit in the present protocol is the lack of a study arm treated with GLP-1 receptor agonist only. Thus, it will not be possible to put together what the single intervention (GLP-1 receptor agonist or SGLT-2 inhibitor) does in comparison to the combination. Questions of additivity/synergism cannot really be addressed with this design. The choice of agents is obviously determined by the sponsor (both exenatide once weekly and dapagliflozin are products marketed by AstraZeneca). Exenatide is not the most potent weight-reducing GLP-1 receptor agonist. As a consequence, a greater sample size is necessary as compared to using a more efficacious GLP-1 receptor agonist. The protocol does not outline the numbers (and the selection) of patients planned to undergo additional experiments for exploratory outcomes. Page 21, line 1: A DMC is mentioned, but its composition or tasks are not further explained.
--

REVIEWER	Muscelli, E
-----------------	-------------

	University of Campinas, Internal medicine
REVIEW RETURNED	22-Nov-2020

GENERAL COMMENTS	This is a very interesting protocol aimed to assess the fat loss induced by a 32-week co-administration of GLP-1 RA (exenatide once weekly) and a SGLT-2 inhibitor (dapagliflozin) to patients with T2D. Many other parameters will be investigated as secondary objectives or exploratory end points, like weight loss, body composition, metabolic parameters, cardiac structure and function, etc. Currently few studies have investigated some of these parameters under this dual concomitant therapy. Their mechanisms of action are independent of each other, thus the association is potentially synergic for some or many of the parameters that will be investigated. The protocol is well planned and state of art methods will be used for the main evaluations. Observations: Introduction  1. I would like to suggest adding some information from previous studies that administered a combination of GLP-1 RA and SGLT-2i drugs. This information would be useful to better justify the implementation of the proposed protocol. 2. Page 8, first line - The attribution of the increased EGP only to the hyperglucagonaemia seems to be a simplification of the EGP response to the SGLT-2i therapy. Perhaps the authors should add that this is one factor among others, such as the decreased insulin secretion. Methods  1. Page 9, line 48 - 52 - please add previous references for choosing 1.8kg of body fat reduction for the sample size calculation. 2. Please include literature references for VAS (appetite evaluation) and for energy expenditure assessment using double labelled water. 3. Page 14, line 42 and page 17 lines 19 and 45 - the number of participants of the subgroups that will be submitted to: - clamps studies; double labelled water; endothelial function tests; cardiac MRI and fMRI is not reported. Did the authors predict how many patients will be included in each investigation? Some abbreviations along the manuscript are not defined, for example ICH GCP, ISDMC, etc. Finally, who will have access to the full dataset after the trial?
---

REVIEWER	Cornu, Catherine Hospices Civils de Lyon, Clinical Investigation Centre
REVIEW RETURNED	25-Jan-2021

GENERAL COMMENTS	On request of the editorial board to improve the reporting of protocols for randomized controlled trials the SPIRIT guidelines were used as reference (Chan et al. SPIRIT 2013 explanation and elaboration: guidance for protocols of clinical trials. BMJ, 2013; 346. PMID: 23303884 Page 8 of 31 : Primary objective of the study is to compare the adjusted mean reduction in total body fat mass (determined by dual-energy Xray absorptiometry, DEXA) from baseline following 32 weeks of treatment with exenatide QW and dapagliflozin versus dapagliflozin alone compared with control.  • the clinical relevance of this outcome should be argued
--

	 • The choice of 32 weeks for measuring the change needs explanation • Absolute mean reduction: is it relative or absolute reduction ? • This not totally in accordance with the analysis presented page 23 where 2 comparisons are presented ((i) control arm vs. dapagliflozin arm, and (ii) dapagliflozin vs. dapagliflozin & exenatide. The alpha risk inflation is not adequately addressed: statistical significance will be tested at an alpha level of 0.05 for each comparison. In the case that only one comparison is significant, then Hochberg’s procedure will be implemented to adjust for multiplicity. Statistical significance would then be inferred using an adjusted alpha level of 0.025. Sample size: P13 : the authors present 15 active patients who are lost to follow-up. How this could introduce a bias, and how it will be manage should be addressed. Page 22-23, the authors state that “Missing data are assumed to be small and analysis will be carried out on a complete case basis. If there are a significant number of missing entries on any endpoint (>20% for example), missing data shall be estimated using multiple imputation based on the method of chained equations”. The authors should specify whether the 15 patients lost to follow-up will be included in the analysis. The y also should present their plans to promote participant retention and complete follow-up, including list of any outcome data to be collected for participants who discontinue or deviate from intervention protocols; this is of particular importance for the primary outcome DEXA. There are many secondary (15) and exploratory outcomes (8), plus safety outcomes . All could be considered exploratory since they will only permit to raise hypotheses. Page 12 of 31: The randomization process needs to be more explicit on how the investigators will actually randomize patients (IVRS, IWRS, other ?), and whether the randomization is adequately concealed. Page 13 of 31: blinding: the authors should specify that “unblinded staff members at site “ have no role in the follow-up or evaluation of patients in the study, and explain how this can be achieved. They should also specify who will be blinded after assignment to interventions (eg, trial participants, care providers, outcome assessors, data analysts) and how. Page 14 of 31: the authors should specify which formula will be used to calculate the estimated glomerular filtration rate (eGFR), and justify this choice. Page 20 of 31: the list of issues in the “Trial monitoring and oversight committees” section is too long, and not detailed enough on processes to promote data quality (e.g., duplicate measurements, training of assessors, etc ...). Page 21 of 31: The role of the DMC could be addressed in more detail. The authors should also indicate where and how the “separate statistical analysis plan” will be consultable. A statement of who will have access to the final trial dataset, and disclosure of contractual agreements that limit such access for investigators should be given; A description of who will have access to the full dataset after the trial and whether individual patient data will be shared in any form with other researchers, the public and patients.
--	--

VERSION 1 – AUTHOR RESPONSE

Reviewer Reports:

Reviewer: 1

Prof. M Nauck, Ruhr-University Bochum

Comments to the Author:

Brown et al. present a protocol for a clinical study comparing dapagliflozin (SGLT-2 inhibitor), exenatide once weekly (GLP-1 receptor agonist) plus dapagliflozin, and placebo over 32 weeks, with the primary endpoint change in fat mass, and various secondary endpoints focusing on energy balance, appetite/satiety and body composition. Mechanistic questions will be examined in subgroups (insulin sensitivity, CNS processing of food-related clues, cardiovascular function, etc.).

1. The question of how a combination of GLP-1 receptor agonists and SGLT-2 inhibitors works, is of high interest, mainly with respect to cardiovascular outcomes (which will not be clarified with the present protocol). We agree that there is great interest in the cardiovascular benefits of these two agents although this is clearly not the main focus of the study. However, as an additional exploratory analysis, a sub-group of participants are invited to have a transthoracic echocardiogram (ECHO) and where possible a cardiac MR (CMR) at baseline and after treatment. We anticipate more than 20 participants per treatment arm will attend for these cardiac investigations and these findings may yield some interesting mechanistic insight into the benefits observed in the cardiovascular outcome studies (CVOTs).

2. The main deficit in the present protocol is the lack of a study arm treated with GLP-

1 receptor agonist only. Thus, it will not be possible to put together what the single intervention (GLP-1 receptor agonist or SGLT-2 inhibitor) does in comparison to the combination. Questions of additivity/synergism cannot really be addressed with this design. The study is not designed to look at the individual effects of each agent but to investigate the compensatory mechanisms that additional GLP-1 RA therapy would have with a SGLT2i considering the potential (mal)adaptations of increased appetite and increased hepatic glucose output with a SGLT2i. We would contend that much is already known regarding the individual benefits of each agent and the excitement/unresolved questions largely relate to the synergy of combination therapy. The feasibility and cost of undertaking a four-arm study is also a major consideration.

3. The choice of agents is obviously determined by the sponsor (both exenatide once weekly and dapagliflozin are products marketed by AstraZeneca). Exenatide is not the most potent weight-reducing GLP-1 receptor agonist. As a consequence, a greater sample size is necessary as compared to using a more efficacious GLP-1 receptor agonist. At the time of protocol development and grant funding being awarded, agents such as subcutaneous or oral semaglutide were not available in this context. Exenatide QW continues to be widely prescribed in clinical practice. There is great excitement and interest in combining a SGLT2i with a GLP-1 RA so it is felt the results of this trial will remain relevant and of great interest.

4. The protocol does not outline the numbers (and the selection) of patients planned to undergo additional experiments for exploratory outcomes. A statement has been added to the 'Exploratory Outcomes' section detailing this (page 16).

5. Page 21, line 1: A DMC is mentioned, but its composition or tasks are not further explained. The section 'Trial monitoring and oversight committees' has been revised to make the role of the ISDMC as well as the Trial Management Group 'TMG' and Trial Steering Committee (TSC) clearer (pages 20-21).

Reviewer: 2

Dr. E Muscelli, University of Campinas

Comments to the Author:

This is a very interesting protocol aimed to assess the fat loss induced by a 32-week co-administration of GLP-1 RA (exenatide once weekly) and a SGLT-2 inhibitor (dapagliflozin) to patients with T2D. Many other parameters will be investigated as secondary objectives or exploratory end points, like weight loss, body composition, metabolic parameters, cardiac structure and function, etc. Currently few studies have investigated some of these parameters under this dual concomitant therapy. Their mechanisms of action are independent of each other, thus the association is potentially synergic for some or many of the parameters that will be investigated. The protocol is well planned and state of art methods will be used for the main evaluations.

Observations:

Introduction

1. I would like to suggest adding some information from previous studies that administered a combination of GLP-1 RA and SGLT-2i drugs. This information would be useful to better justify the implementation of the proposed protocol. We agree this is important. We have signposted the reader to existing studies using this

combination and the overall additive benefits of glycaemic improvement and weight loss in the 'Background and Rationale' section of the study (page 6).

2. Page 8, first line - The attribution of the increased EGP only to the hyperglucagonaemia seems to be a simplification of the EGP response to the SGLT- 2i therapy. Perhaps the authors should add that this is one factor among others, such as the decreased insulin secretion. You are right to highlight this. In response we have made changes to the text (page 6).

Methods

1. Page 9, line 48 - 52 - please add previous references for choosing 1.8kg of body fat reduction for the sample size calculation. A variety of the licensing studies with dapagliflozin had suggested an average weight loss of ~2.5 kg. This data is summarised in a meta-analysis published around this time (Zhang et al, 2014). Also based on the study by Bolinder et al, it was demonstrated that approximately 2/3 of weight loss is accounted for by reductions in fat mass. The investigators used DEXA to measure body composition as we have done. From this reduction in total body weight of 2.5 kg and 2/3 of total body weight being reduction in fat mass, we derived the predicted 1.8kg reduction in fat mass value.

Zhang Q, Dou J, Lu J. Combinational therapy with metformin and sodium-glucose cotransporter inhibitors in management of type 2 diabetes: systematic review and meta-analyses. *Diabetes Res Clin Pract.* 2014;105(3):313–21.

Bolinder J, Ljunggren Ö, Kullberg J et al. Effects of Dapagliflozin on Body Weight, Total Fat Mass, and Regional Adipose Tissue Distribution in Patients with Type 2 Diabetes Mellitus with Inadequate Glycemic Control on Metformin. *J Clin Endocrinol Metab*, 2012 Mar;97(3):1020-31.

2. Please include literature references for VAS (appetite evaluation) and for energy expenditure assessment using double labelled water.

VAS: Blundell et al, 2010

Blundell J, C de Graaf, Hulshof T et al. *Obes Rev*, 2010;11(3):251-70. doi: 10.1111/j.1467-789X.2010.00714.x.

DLW: Sanghvi et al, 2015

Sanghvi A, Redman LM, Martin CK, Ravussin E, Hall KD. Validation of an inexpensive and accurate mathematical method to measure long-term changes in free-living energy intake. *The American journal of clinical nutrition*. 2015 Aug 1;102(2):353-8.

3. Page 14, line 42 and page 17 lines 19 and 45 - the number of participants of the subgroups that will be submitted to: - clamps studies; double labelled water; endothelial function tests; cardiac MRI and fMRI is not reported. Did the authors predict how many patients will be included in each investigation? A statement has been added to the 'Exploratory Outcomes' section detailing this (page 16).

4. Some abbreviations along the manuscript are not defined, for example ICH GCP, ISDMC, etc. Where these terms remain in the manuscript they have been defined in full.

5. Finally, who will have access to the full dataset after the trial? A data availability statement has been added to the manuscript (page 27).

Reviewer: 3

Dr. Catherine Cornu, Hospices Civils de Lyon Comments to the Author:

On request of the editorial board to improve the reporting of protocols for randomized controlled trials the SPIRIT guidelines were used as reference (Chan et al. SPIRIT 2013

explanation and elaboration: guidance for protocols of clinical trials. *BMJ*, 2013; 346. PMID: 23303884

Page 8 of 31 : Primary objective of the study is to compare the adjusted mean reduction in total body fat mass (determined by dual-energy Xray absorptiometry, DEXA) from baseline following 32 weeks of treatment with exenatide QW and dapagliflozin versus dapagliflozin alone compared with control.

1. The clinical relevance of this outcome should be argued. The primary outcome was chosen based on well-established knowledge of anticipated reductions in weight and HbA1c. Fat mass was considered a relevant outcome measure to the mechanisms being studied (relating to appetite and other measures of body composition).

2. The choice of 32 weeks for measuring the change needs explanation. Existing trials combining a SGLT2i and GLP-1 RA were performed over 24-32 weeks. In this trial 32 weeks was deemed appropriate to examine the short- and long-term effects of this combination on body composition and appetite etc.

3. Absolute mean reduction: is it relative or absolute reduction ? Analysis will be undertaken using Analysis of Covariance which measures the absolute reduction whilst adjusting for baseline values. The statistical analysis plan does however allow for transformation of the primary outcome based on the residual diagnostics and if a transformation is required (e.g. log) transformation then analysis would be on the relative as opposed to absolute scale.

4. This not totally in accordance with the analysis presented page 23 where 2 comparisons are presented (i) control arm vs. dapagliflozin arm, and (ii) dapagliflozin vs. dapagliflozin & exenatide. The

alpha risk inflation is not adequately addressed: statistical significance will be tested at an alpha level of 0.05 for each comparison. In the case that only one comparison is significant, then

Hochberg's procedure will be implemented to adjust for multiplicity. Statistical significance would then be inferred using an adjusted alpha level of 0.025. Thank- you for identifying the discrepancy. Statistical significance will be assessed using the $P < 0.025$ level to protect the global type 1 error rate < 0.05 .

5. Sample size: P13 : the authors present 15 active patients who are lost to follow-up.

How this could introduce a bias, and how it will be managed should be addressed. Page 22-23, the authors state that "Missing data are assumed to be small and analysis will be carried out on a complete case basis. If there are a significant number of missing entries on any endpoint ($>20\%$ for example), missing data shall be estimated using multiple imputation based on the method of chained equations". The authors should specify whether the 15 patients lost to follow-up will be included in the analysis. They also should present their plans to promote participant retention and complete follow-up, including list of any outcome data to be collected for participants who discontinue or deviate from intervention protocols; this is of particular importance for the primary outcome DEXA. Thank you for this. The 15 patients lost-to-follow-up will not be included in the final analysis as they do not provide primary outcome data. No imputation is planned as described in these missing patients, as they do not meet the 20% threshold suggested in the Statistical Analysis Plan. Patterns of missingness are summarised as part of the statistical analysis plan to investigate any association with the study outcome of treatment allocation. A Sensitivity analysis performed using multiple imputation is also planned.

To promote retention and complete follow-up, each participant will be asked to complete a patient diary which records anthropometric measurements, capillary blood sugars and medication compliance. The diary will be used to facilitate

discussion at each study visit and answer any concerns the participant might have ('Dosage and administration of study treatments', page 13).

6. There are many secondary (15) and exploratory outcomes (8), plus safety outcomes.

All could be considered exploratory since they will only permit to raise hypotheses. We have separated these endpoints in this way, firstly to help the reader but specifically because all participants will have measurements taken for the secondary endpoints but only a sub-group of participants will have the exploratory endpoint measurements for reasons we have detailed.

7. Page 12 of 31: The randomization process needs to be more explicit on how the investigators will actually randomize patients (IVRS, IWRS, other ?), and whether the randomization is adequately concealed. We have updated the 'screening, enrolment and randomisation' section in order to answer this (page 12). Information in the 'Dosage and administration of study treatments' section is also useful for information on concealment of treatment Arm.

8. Page 13 of 31: blinding: the authors should specify that "unblinded staff members at site " have no role in the follow-up or evaluation of patients in the study, and explain how this can be achieved. They should also specify who will be blinded after assignment to interventions (e.g., trial participants, care providers, outcome assessors, data analysts) and how. Please see response to question 7 which partially answers this questions. The role of unblinded members of the research team has been highlighted in the 'Monitoring/dispensing visits' section (page 19).

9. Page 14 of 31: the authors should specify which formula will be used to calculate the estimated glomerular filtration rate (eGFR), and justify this choice. Routine biochemistry is performed in the local Liverpool Clinical Laboratories. The Modification of Diet in Renal Disease (MDRD) formula is the only formula

currently used to estimate glomerular filtration rate (eGFR) in this lab. We have specified which formula is used in the manuscript (page 13).

10. Page 20 of 31: the list of issues in the “Trial monitoring and oversight committees” section is too long, and not detailed enough on processes to promote data quality (e.g., duplicate measurements, training of assessors, etc ...). The section ‘Trial monitoring and oversight committees’ has been revised to make the role of the ISDMC as well as the Trial Management Group (TMG) and Trial Steering Committee (TSC) clearer (pages 20-21).

11. Page 21 of 31: The role of the DMC could be addressed in more detail. The authors should also indicate where and how the “separate statistical analysis plan” will be consultable. The section ‘Trial monitoring and oversight committees’ has been revised to make the role of the Independent Safety and Data Monitoring Committee (ISDMC) as well as the Trial Management Group (TMG) and Trial Steering Committee (TSC) clearer (pages 20-21).

12. A statement of who will have access to the final trial dataset, and disclosure of contractual agreements that limit such access for investigators should be given; A description of who will have access to the full dataset after the trial and whether individual patient data will be shared in any form with other researchers, the public and patients. A data availability statement has been added at the end of the manuscript (page 27).

VERSION 2 – REVIEW

REVIEWER	Cornu, Catherine Hospices Civils de Lyon, Clinical Investigation Centre
REVIEW RETURNED	17-Mar-2021

GENERAL COMMENTS	The revised version addresses most of the comments, however it is still not totally clear about the alpha risk management: since there are two primary comparisons, is there an adjustment of the alpha risk or a hierarchical analysis of the primary outcome ?
--

REVIEWER	Muscelli, E University of Campinas, Internal medicine
REVIEW RETURNED	31-Mar-2021

GENERAL COMMENTS	I consider that the authors addressed the comments. However, the references of DEXA, VAS and DLW should be included in the manuscript.
--

REVIEWER	Nauck, M Ruhr-University Bochum, Division of Diabetology
REVIEW RETURNED	02-Apr-2021

GENERAL COMMENTS	I still believe that the weaknesses identified in the first round of reviews will limit the conclusions that can be derived from this (versus a potentially amended) protocol. This needs to be discussed once the data are there and will be published.
--

VERSION 2 – AUTHOR RESPONSE

Reviewer Reports:

Reviewer: 3

Dr. Catherine Cornu, Hospices Civils de Lyon Comments to the Author:

The revised version addresses most of the comments, however it is still not totally clear about the alpha risk management: since there are two primary comparisons, is there an adjustment of the alpha risk or a hierarchical analysis of the primary outcome

The lead trial statistician has reviewed this valid comment and made the following change to the manuscript:

“For the analysis of the primary outcome, statistical significance will be tested using Hochberg’s procedure, a hierarchical analysis approach for the primary outcomes of (i) exenatide QW and dapagliflozin compared to placebo control, (ii) dapagliflozin alone compared with placebo control. Here comparisons will be ordered (descending) in terms of their significance level and assessed using a p-value <0.05 to determine statistical significance for the first comparison and p<0.025 for the second.”

Reviewer: 2

Dr. E Muscelli, University of Campinas Comments to the Author:

I consider that the authors addressed the comments.

However, the references of DEXA, VAS and DLW should be included in the manuscript. We introduce the abbreviations as they appear in the text and tables. If you would like a list of abbreviations to appear in the publication we would suggest the following: AE, adverse event; BIA, bio-electrical impedance analysis; BP, blood pressure; DEXA, dual-energy X-ray absorptiometry; DLW, doubly labelled water; eGFR, estimated glomerular filtration rate; ECG, electrocardiogram; FMD, flow mediated dilatation; GLP-1 RA, glucagon-like peptide-1 receptor agonist; ISDMC, Independent Safety and Data Monitoring Committee; LCTC, Liverpool Clinical Trials Centre; METS, metabolic equivalents; MRI, magnetic resonance imaging; fMRI, functional MRI; SGLT2i, sodium glucose co-transporter 2 inhibitor; TMG, Trial Management Group; TSC, Trial Steering Committee; T2D, Type 2 diabetes;

VAS, visual analogue scale.

Reviewer: 1

Prof. M Nauck, Ruhr-University Bochum Comments to the Author:

I still believe that the weaknesses identified in the first round of reviews will limit the conclusions that can be derived from this (versus a potentially amended) protocol.

This needs to be discussed once the data are there and will be published. We hope to publish the findings of this paper in high impact journals. It is anticipated the results will stimulate useful discussion in the field.